# The Social Stratification of Availability, Affordability, and Consumption of Food in Families with Preschoolers in Addis Ababa; The EAT Addis Study in Ethiopia

**DOI:** 10.3390/nu12103168

**Published:** 2020-10-16

**Authors:** Semira Abdelmenan, Hanna Y. Berhane, Magnus Jirström, Jill Trenholm, Alemayehu Worku, Eva-Charlotte Ekström, Yemane Berhane

**Affiliations:** 1Addis Continental Institute of Public Health, 26751/1000 Addis Ababa, Ethiopia; hannayaciph@gmail.com (H.Y.B.); alemayehuwy@yahoo.com (A.W.); Lotta.Ekstrom@kbh.uu.se (E.-C.E.); yemaneberhane@gmail.com (Y.B.); 2Department of Women’s and Children Health, International Maternal and Child Health, Uppsala University, 751 85 Uppsala, Sweden; jill.trenholm@kbh.uu.se; 3Institute of Public Health, College of Medicine and Health Sciences, University of Gondar, 196 Gondar, Ethiopia; 4Department of Human Geography, Lund University, 223 62 Lund, Sweden; magnus.jirstrom@keg.lu.se; 5School of Public Health, College of Health Sciences, Addis Ababa University, 1176 Addis Ababa, Ethiopia

**Keywords:** social stratification, dietary diversity, availability, affordability, food environment, Ethiopia

## Abstract

The aim of this study was to understand the quality of diet being consumed among families in Addis Ababa, and to what extent social stratification and perceptions of availability and affordability affect healthy food consumption. Data were collected from 5467 households in a face-to-face interview with mothers/caretakers and analyzed using mixed effect logistic regression models. All family food groups, except fish were perceived to be available by more than 90% of the participants. The food groups cereals/nuts/seeds, other vegetables, and legumes were considered highly affordable (80%) and were the most consumed (>75%). Households with the least educated mothers and those in the lowest wealth quintile had the lowest perception of affordability and also consumption. Consumption of foods rich in micronutrients and animal sources were significantly higher among households with higher perceived affordability, the highest wealth quintile, and with mothers who had better education. Households in Addis Ababa were generally seen to have a monotonous diet, despite the high perceived availability of different food groups within the food environment. There is a considerable difference in consumption of nutrient-rich foods across social strata, hence the cities food policies need to account for social differences in order to improve the nutritional status of the community.

## 1. Introduction

Malnutrition is one of the most serious public health challenges globally; nearly one in three people suffer from at least one form of malnutrition manifested as wasting, stunting, micro-nutrient deficiency, and/or overweight or obesity [1]. In the past two decades, the number of overweight individuals in Africa has significantly increased. This is a major concern especially in light of the remaining high prevalence of undernutrition in this region [2,3]. The occurrence of both under-and over nutrition in the same population is often referred to as the “double burden of malnutrition” [4]. The driving forces are multifaceted, including changes in dietary and physical activity patterns related to transitions in livelihoods; moving from traditional food systems dependent upon subsistence farming to dependence upon income generation and the market [1]. These transitions are more prominent in transient urban settings where they contribute to an excess intake of energy, insufficient amounts of micro-nutrients and limited physical activity [5,6,7].

Despite the various health benefits of fruits and vegetables including reducing risk for non-communicable diseases, the consumption is generally low [8]. An aggregated result from 28 low-and middle-income countries exemplified this with only 18% (95 CI: 16.6–19.4%) of the population meeting the recommended 400 g/day serving [9]. Traditional diets rich in fiber, high in vegetable and fruit varieties are being substituted by diets rich in oils, fats, and sugars, often in the form of ultra-processed foods [10,11]. The shift in diet can be partially explained by changes in food systems; whereby the rise of industrial production of cheap foods is making ultra-processed food more accessible [12,13,14].

With regards to food accessibility, the importance of the “food environment” is underscored [13,15,16]. The food environment which encompasses domains such as accessibility, availability, and affordability is an interface whereby individuals interact with the wider food system for food purchase and consumption [17]. Logically, individuals can only eat from the range of things that are available to them, and availability of an adequate supply of healthy food has been consistently associated with a healthy diet [18]. The availability of fresh healthy foods has also been improved by the development of advanced agricultural technologies and transport systems that can provide cold chain equipment to preserve perishables. However, as these technologies are undeveloped in low income settings 40% of fresh food does not reach consumers due to post-harvest loss [19,20].

Another important dimension relating to diet is affordability [18]. In low income urban areas where availability is limited, only the economically better-off families manage to access a wide range of the recommended food groups [20,21]. Despite sufficient access to markets, dietary choices are influenced by price [22]. The results from a multi-country study showed that individuals from low income countries spend more than half of their income in order to meet dietary recommendations [23]. The high cost prevents poorer households from affording a nutritionally adequate diet [22]. This is more common in urban areas where most households’ only source of food is that which is purchased. Furthermore, less healthy alternatives are often inexpensive, have a longer shelf-life, require little preparation, and have an alluring taste, making them both convenient and desirable for the women who are responsible for purchasing and preparing food for the family [13,24,25].

Mothers/caretakers have a significant role in shaping the diet of the household as they generally determine which foods are bought and prepared within the home. Mothers draw on wealth and food security of the household as well as their own education to fulfill this responsibility. Studies have shown that households with better socioeconomic resources enjoy a more nutritious diet, as opposed to their disadvantaged counterparts whose diet consists of much less fiber, fruits, and vegetables [26,27,28]. Households with educated mothers have been shown to have a higher consumption of fruits and vegetables as well as better diet diversity [9,29,30]. On the contrary, in some contexts, higher maternal education has also been associated with higher consumption of sugary drinks and more processed foods [31]. Moreover, although households in the highest wealth groups had a more diverse diet [26,32], they have also been associated with an increase in overweight/obesity [33,34].

The fundamental cause of the double burden of malnutrition is energy imbalance; “energy excess” for obesity and “energy and nutrient deficiencies” for undernutrition [35]. Hence the promotion of good nutrition presents an opportunity to avert the occurrence of both forms of malnutrition [34]. However, there is limited research, in particular concerning the food environments in sub-Saharan African countries, which makes designing holistic and contextualized policies problematic. Therefore, understanding how availability and affordability shapes the family diet is important. This study aims to understand the quality of diet being consumed among families in Addis Ababa, and to what extent social stratification and perceptions of availability and affordability affect healthy food consumption.

## 2. Materials and Methods

### 2.1. Study Design and Setting

A community-based cross-sectional study was carried out in the months of July–August, 2017 and January–February, 2018 in Addis Ababa, the capital city of Ethiopia. Addis Ababa has been experiencing a rapid increase in population size along with diminishing open public space [36]. The rapid expansion of residential areas to accommodate the increasing population size has led to the loss of highly fertile agricultural land and green spaces thereby reducing food production within and in the vicinity of the city [37]. This in turn, escalates the food prices and further jeopardizes the food and nutrition security of urban dwellers. Additionally, the city has one of the highest literacy rates in the country with 80% of its population having basic literacy level, and a high level of unemployment with 23.5% of the population in this urban area being unemployed [38,39].

### 2.2. Sampling

The study used a multi-stage sampling procedure. All 116 woredas (districts) in the city were included in the study in each of the two rounds of survey: First round took place during the wet season and second round during the dry season to consider seasonal variations. Each district was further divided into roughly five equal geographical clusters and one cluster was then selected using simple random sampling. Subsequently, systematic random sampling was used to visit 60 households in each cluster to check for eligibility. All households with at least one under five-year-old child were invited to participate. Additional households were visited if there were less than 20 eligible households in the cluster. Households in which the mother/caregiver was not available to interview after 3 repeat visits were then declared unavailable and excluded from the study without replacement.

### 2.3. Data Collection

The necessary data for the study were collected through face to face interviews with the mother/caregiver using a structured questionnaire. The questionnaires included sections on demographic and household characteristics, perceived availability and affordability of food groups, and family food consumption. The questionnaire comprised of standard measures as well as newly developed measures to assess the perceived affordability and availability; this was based upon literature and the research team’s expertise in this field. A photo gallery of common foods was used to help respondents understand the food groups. The questionnaires were initially developed in English and translated into Amharic language (the official language of Ethiopia) by a panel of translators. The questionnaires were pretested in households that were not included in the study for comprehension of concepts and language.

Data were collected using tablets pre-programmed with the questionnaire. The data collectors were trained on the objective of the study, the content of the questionnaires, the sampling procedures, and the use of the tablets. The data collection was supervised daily by members of the research team and on-site support was given to the teams to ensure the study procedures were strictly observed. Data were received directly onto the data server at the Addis Continental Institute of Public Health and a data manager provided regular feedback on the quality of data.

### 2.4. Measurement

The age of the mothers was grouped into five categories: 15–24, 25–34, 35–44, and 45+, and their educational levels were summarized as: never attended/finished a grade, grade 1–4, grade 5–8, grade 9–12, and college. Marital status was categorized as currently married (in union) and currently not married (single).

Wealth index was computed from multiple variables including ownership of house, type of housing unit, housing material (floor, roof, wall material), access to a separate toilet facility and clean drinking water, and assets (including bicycle, motorbike, car, cell-phone, radio, TV, refrigerator, bed, electric stove for making the local bread “Injera” and a saving account) [40]. Households were then divided into wealth quintiles (lowest, second, third, fourth, and highest) to indicate their relative economic status.

#### Consumption, Availability, and Affordability of Food Groups

For this study, a modified version of the women’s minimum dietary diversity indicator was used; it measures quality of diet, both in terms of energy and micronutrient adequacy [41] instead of using the usual household diet diversity measures which reflect more on the economic access and dietary energy [42]. This modified measure used in this study, hereafter referred to “family food groups”, has eleven food groups rather than 10 as in the usual measures. Based on the local consumption patterns; food groups “fish and meat” and “Vitamin A rich fruits and vegetables” were both split; while merging “legumes” with “nuts and seeds” groups since the latter is not commonly consumed in the study setting.

Perceived availability of family food was measured using a photo gallery of common foods from each of the family food groups (11-family food groups). Mothers were asked whether any of the foods shown in the photo were available in the market. The response options were “Yes”, “No”, and “Don’t know”. Then, each food group was dichotomized as “available” if the responses were “yes” and “not available” if the responses were “no”. “Don’t know” responses were treated as missing.

Perceived affordability of family food was assessed by asking the mother/caregiver, “How often can your family afford to consume any of these foods?” Response options were coded: “as often as wanted”, “a little less frequently than wanted”, and “much less frequently than wanted/not at all”. Perception of affordability was dichotomized as “affordable” if response was “as often as wanted” and “not affordable” for the other categories.

Household food consumption was measured by a combination of two complimentary methods; first the mothers were asked to recall foods consumed by the family in the last 24 h. Then, the enumerator read and showed the photos of common foods by family food group whilst asking: “did any household member consume any of these foods in this photo in the last 24 h?” The response options included “yes”, “no”, and “don’t know”; the response category “don’t know” was treated as missing.

### 2.5. Statistical Analysis

Analysis was conducted using the statistical software program Stata version 15.0 [43]. Standard descriptive statistics were computed for outcome and explanatory variables including percentages and their respective confidence intervals for categorical variables as well as mean and standard deviation for continuous variables. Mixed effect logistic regression models were used to assess the bivariate and multivariable associations between the dependent variables’ family food consumption and the explanatory variables including perceived affordability, wealth quintiles, and maternal education. All models were adjusted for clustering at district level. *p*-values of <0.05 were considered as statistically significant. The variance of random effect value along with 95% confidence intervals (CI) and standard error (SE) computed to observe heterogeneity between districts. An additional intraclass correlation coefficient (ICC) was conducted to check variance at district level.

### 2.6. Ethical Consideration

Ethical approval for the EAT Addis study was obtained from the institutional review board of Addis Continental Institute of Public Health with reference number ACIPH/IRB/004/2015 and University of Gondar institutional review board reference number V/P/RCS/05/352/2019. Verbal informed consent was obtained from each of the participants after explaining the purpose of the study and addressing any questions. Permission to conduct the study was obtained from all sub-cities and district level health offices.

## 3. Results

A total of 14,018 households were visited to identify eligible households, 8293 households had no children under the age of five years and in 194 of the households, respondent mothers/caregivers were not available. Finally, with the response rate of 98.8%, a total of 5467 household were included in the study (Figure 1).

More than half of the mothers (61.1%) were in the age group, 24–34 years, 88% were married and 47% had 12 or more years of schooling. The average family size for households in our study was 4.3 ± 1.5, and only about one-fifth of families owned the house they live in (21.3%) (Table 1).

### 3.1. Perceived Availability, Affordability, and Consumption of Family Food

Most food groups were perceived to be highly available in the market (Table 2). All food groups were reported to be available by more than 90% of the participants except for fish (73.5%). Due to the reported high perceived availability of the food items, the social stratification and its explanatory contribution to household consumption was not reported.

Considerable variations were observed in the perceived affordability of family foods ranging from 16% to 89%. Fish was perceived affordable only by 16.2% (95% CI: 15.2, 17.2) of the households. Other food groups were perceived to be affordable by less than 50% of households; vitamin A rich fruits (38.3%; 95% CI: 37.0, 39.6), other fruits (46.6%; 95%CI: 45.3, 48.0), and meat (32.4%; 95% CI: 31.2, 33.7). The highly perceived affordable food groups included cereals and white roots/tubers (89.2%; 95% CI: 88.4, 90.1), other vegetables (86.0%; 95% CI: 85.1, 87.0), and legumes, nuts and seeds (84.5%, 95% CI: 83.5, 85.4) (Table 2).

The main foods consumed in the study population were cereals and white roots/tubers (99.3%; 95% CI: 99.0, 99.5), followed by other vegetables (97.5; 95% CI: 97.1, 97.9), and legumes, nuts and seeds (84.9%; 95% CI: 84.0, 85.9). The reported consumption of fish was the lowest (2.3%; 95% CI: 1.9, 2.7). The consumption of other food groups was also reported by 50% or less proportion of the households; vitamin A rich fruits (10.0%; 95% CI: 9.2, 10.8), dark green leafy vegetables (23.1%; 95% CI: 22.0, 24.2), vitamin A rich vegetables (24.9%; 95% CI: 23.7, 26.0), other fruits (33.2%; 95% CI: 32.0, 34.5), meat (31.5%%; 95% CI: 30.3, 32.8), eggs (27.7%%; 95% CI: 26.5, 28.9), and dairy products (45.9%; 95% CI: 44.5, 47.2). The consumption pattern showed households have a monotonous staple diet with limited diet diversity (Table 2). The three most highly consumed food groups i.e., cereals and white roots/tubers, other vegetables, and legumes, nuts and seeds were not considered in the regression models evaluating associations with wealth and maternal education.

### 3.2. Social Stratification of Perceived Affordability and Consumption of Family Food

The perceived affordability and consumption showed significant social stratification by wealth and maternal education; the least educated caretakers and the poorest quintile had the lowest perception of affordability as well as consumption for most of the food groups. Both affordability and consumption showed significant differences between the lowest and the highest wealth quintiles and maternal education categories for all food groups except cereals/white roots and tubers. The largest differences in perceived affordability by wealth (lowest versus highest quintile) and education (no education versus college) was observed in fish (3.9-fold by wealth and 4.4-fold by education), meat (3.4-fold by wealth and 3.9-fold by education), and vitamin A fruits (3-fold by wealth and 3.4-fold by education). Similarly, the three largest differences in consumption were observed for fish (7.9-fold by wealth and 6-fold by education), vitamin A rich fruits (3.6-folds by wealth and 5.1-folds by education), and meat (3.1-fold by wealth and 2.7-fold by education) (Table 3).

### 3.3. Family Food Consumption and Association Attributed to Individual Indicators

To evaluate the relative contribution of wealth, maternal education, and perceived affordability to family consumption of different food groups, we examined their independent association.

After controlling for wealth, maternal education, food security status, and clustering effect in the model, those who reported high affordability (could purchase as much as wanted) were more likely to consume the food items. Except for vitamin A rich fruits, the odds of consuming micronutrient rich foods (vitamin A rich vegetables, dark green leafy vegetables, and other fruits) and protein rich animal source foods (meat, dairy, and eggs) were significantly higher in households that perceived they could afford as much as they wanted as compared to those who could not afford (“not at all/afford much less than I want” categories) (Table 4).

Regarding wealth, after adjusting for maternal education, affordability, food security, and cluster, households in the top wealth categories i.e., fourth and highest quintiles had higher odds of consuming micronutrient rich foods (vitamin A rich fruits and vegetables, dark green leafy vegetables and other fruits) compared to households in the lowest quintile. The odds of households in the highest wealth quintile to consume protein rich animal source foods (meat, dairy, and eggs) were more than 2 times higher than those in the lowest wealth quintile (Table 5).

Regarding maternal education, the adjusted analysis revealed that households with college educated mothers had higher odds for consuming micronutrient rich foods (vitamin A rich fruits and vegetables, dark green leafy vegetables, and other fruits) as compared to households with the least educated mothers (never attended/finished a grade). Consumption of Vitamin A rich fruits showed the highest variation; households with college educated mothers were about 4 times more likely to consume them compared to household with mothers who never attended/finished a grade (AOR: 3.77; 95%CI: 2.43, 5.85). When compared with households that had mothers who never attended/finished a grade; those with a college educated mother had about 2 times higher odds of consuming protein rich animal source foods, meat (AOR: 2.1; 95%CI: 1.64–2.69), eggs (AOR: 2.51; 95%CI: 1.93–3.26) and dairy products (AOR: 2.43; 95%CI: 1.96–3.01) (Table 6).

As indicated in Table 7, considerable heterogeneity was not observed between the study clusters. The ICC showed that from 2% to 7% of the variation attributes to cluster level variance. The lowest variance was observed in dairy, Vitamin A rich and dark green vegetables consumption while the highest variance was observed in meat consumption. (Table 7).

## 4. Discussion

This study found that all family food groups were perceived to be available in the market. A few food groups (cereals, other vegetables, and legumes) were perceived as affordable by the majority and were also consumed to a large extent. On the contrary, micronutrient rich foods (vitamin A rich fruits and vegetables and dark green leafy vegetables) as well as animal source protein rich foods (meat, dairy, and eggs) showed a considerable variation in perceived affordability as well as consumption with a clear stratification in terms of household wealth and maternal education. Evaluation of the relative contribution of affordability, wealth, and maternal education to consumption of family foods showed, with a few exceptions, that all three of them independently explained consumption of the various family foods; higher perceptions of affordability and socioeconomic resources were associated with higher prevalence of consumption.

All of the family food types with the exception of fish were reported to be available by 90% or more of the respondents. This gives a clear indication that most foods required for a healthy diet are perceived to be available for the vast majority of households. The high availability of foods may be explained by the diversities in food retail alternatives and a high dependability on informal markets which makes the food market more responsive to the need of vulnerable segments of the population [44,45]. This, in turn, makes the more vulnerable less susceptible to food deserts which often occur in urban high-income settings with a high dependence on supermarkets [45]. Another explanation for high availability could be that the supply is increasingly following the demand of the population; as the population size increases, the produce coming to the capital may be increasing. This is especially true for cereals/white roots/tubers which constitutes a major part of the staple diet [46].

While a social stratification was shown, the three food groups (cereals/white roots and tubers, legumes, and other vegetables) were considered affordable by a large proportion of the households as well as were consumed by almost all. Across all social strata, consumption of these three types of foods was the same or higher than what households reported to be able to afford. On the contrary, for the other food groups, households consumed less than what they can afford. This could be due to the interplay of many other factors such as preferences, ability to prepare the foods in the family food groups in terms of having adequate cooking space, time, and skills, all of which have been identified as potential drivers for food choices [13,47,48,49].

The high consumption of cereals and white roots/tubers observed here is consistent with national food consumption survey which reported more than 65% of the energy intake of the population is from carbohydrates [46]. Part of the explanation for the affordability of cereals could be high production [50]; the surplus production could potentially make it slightly more affordable than other items in the market. These perceptions align to the realities of the cost of food in Ethiopia [51]. Except for cereals and white roots/tubers, the price of all other food group categories, particularly vitamin A rich dark green leafy vegetables and animal source food, are increasing markedly [51]. Part of the increase in price of perishable foods could be due to infrastructure and transportation systems resulting in large post-harvest losses [20]. This may also push the price of fruits and vegetables higher. As was reflected earlier, affordability rather than availability is a more serious challenge in improving the family diet in urban settings [21]. Except for dark green leafy vegetables, both micronutrient rich foods (vitamin A rich fruits and vegetables) as well as protein rich foods (meat, dairy and eggs) were neither perceived affordable nor actually consumed by the majority of the households. The high cost of a nutritious diet makes the poor more vulnerable; for example, in low income countries 29% of a household’s income is required to purchase one serving of fruits [23]. In another study where they estimated the cost of a recommended healthy diet relative to households per capita, they found that the cost exceeded the total income for 20% of the world’s populations [52], making it impossible for poor households to meet dietary recommendations.

Households in the highest wealth quintile had higher odds of consuming both micronutrient rich foods (vitamin A rich fruits and vegetables and dark green leafy vegetables) as well as protein rich foods (meat, dairy, and eggs). The difference in consumption between households in the highest compared to the lowest wealth quintile was nearly double for both micronutrient rich as well as animal source protein rich foods and remained significant after adjusting for known confounding factors. Other studies have also shown that higher household wealth status significantly improved purchase and consumption of a diverse diet [26,53]. The better-off households were less vulnerable to price fluctuations putting them in a better position for a more planned diet than the poor households [5,47].

Perceptions of affordability were associated with higher odds of consumption of the food groups. This is in line with studies on fruits and vegetables that have shown these food groups were perceived expensive and hence consumed in a limited amount particularly among the disadvantaged groups [26,54]. Individuals from the lowest wealth quintile may not have adequate cold storage options such as refrigerators to store short-lived items such as meat, dairy, and even fruit and vegetable leading to purchases for one-time use, thus paying higher prices to retailers, which renders these items more unaffordable [55]. Food items that were considered affordable, were widely consumed by our study groups indicating cost is a strong determinant for their diets [22,56].

Maternal education level was also associated with better consumption of a variety of food groups. In households with college educated mothers, consumption of both micronutrient and animal source food was higher compared to households with mothers who never attended/finished a grade. Studies have shown that mothers’ awareness of dietary recommendations and nutrition knowledge as well as diet-related diseases are important factors in diet quality [29]. Understanding the health value of food items is linked to years of schooling; higher levels of schooling have been associated with improved diet diversity [57]. Another possible explanation for the improvement in consumption of different food groups is the fact that educated mothers, are often working outside home, have more self-efficacy in making decisions and subsequently a better income, all of which have been shown to improve the diet of the family [30,58].

This study is the largest survey in Addis Ababa on the subject matter and draws a sample from all 116 districts (the smallest administrative unit) of the city. To cover seasonal variations due to the availability of food items, the survey was conducted in two rounds; in what is referred to as the season of plenty and the season of scarcity. The response rate was very high in almost all districts. Thus, the findings are a fair representation of the situation in Addis Ababa. That said, cautious interpretation is advised when attempting to generalize the results to other contexts as Addis Ababa presents a slightly different demographic compared to other cities within Ethiopia, due to its diverse population drawing/migrating from all corners of the country.

For this study we developed two new measures: perceived availability and perceived affordability of foods. The translation of the concepts into Amharic were discussed with native speakers in and outside of the research team before pilot testing. In the pilot testing we found that the concepts were easily understood. Preliminary comparison between perceived availability and our simultaneously collected data on actual availability of foods in the residential neighborhoods show good agreement. Further, preliminary analyses show that perceptions of affordability have a socio-economic stratification in the way it is expected. However, while perception-based measurements provide an understanding of people’s perceptions it is difficult to know what their imbedded dimension are. We suggest further research on this.

We used a standard method for sampling of households. As the sampling is based on selection of a physical dwelling (even if at times of low standard) it does exclude homeless inhabitants of Addis Ababa, unfortunately this is almost unavoidable. Finally, the limitations are as always, a risk of potential social desirability in responses as well as residual confounding in statistical analyses.

## 5. Conclusions

In conclusion, although all the food groups necessary for leading a healthy and nutritious diet were perceived to be available in the food environment, there was a large variation in the perceived affordability and in the actual consumption among the households of Addis Ababa. Generally, households’ food consumption lacked diversity, especially the consumption of micronutrient-rich and animal-source protein-rich foods which were highly limited. All three socio-economic aspects: perception of affordability, maternal education, and household wealth were independently associated and to a similar extent, contributed to variations in consumption. Hence, food policy in this rapidly urbanizing city needs to consider these factors in order to improve the nutritional status of children and families. With cautious interpretation these recommendations may be useful to other in low-income countries’ major cities.

## Figures and Tables

**Figure 1 nutrients-12-03168-f001:**
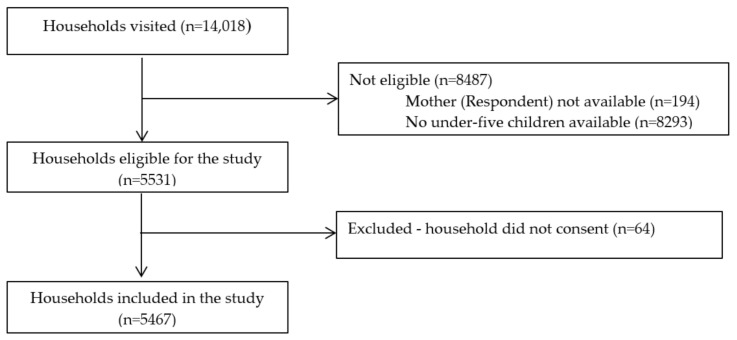
Participant flowchart of households included in the study, Addis Ababa, Ethiopia.

**Table 1 nutrients-12-03168-t001:** Household, and women’s characteristics, Addis Ababa, Ethiopia.

Level	Characteristics (n = 5467)	% (n) or Mean (±SD)
**Women**	Age	
	15–24	15.8 (864)
	25–34	61.1 (3342)
	35–44	18.3 (999)
	45 and above	4.8 (262)
	Education	
	Never attend/Not finished first grade	13.8 (752)
	Grade 1–4	9.1 (498)
	Grade 5–8	30.0 (1638)
	Grade 9–12	27.1 (1482)
	College	20.1 (1097)
	Marital status (Married)	88.0 (4813)
	Involvement in income earning activity	26.2 (1432)
**Household**	Male headed households	86.5 (4, 729)
	Owns a house	21.3 (1165)
	Family size	4.3 (±1.5)
	Access to improve water	98.5 (5386)
	Toilet facility	
	No facility	1.4 (79)
	Shared facility	76.1 (4160)
	Private facility	22.5 (1228)

**Table 2 nutrients-12-03168-t002:** Perceived availability, affordability and consumption of family foods in Addis Ababa, Ethiopia.

Food Group	Perceived Availability		Perceived Affordability		Consumption	
	n/N	% (95% CI)	n/N	% (95% CI)	n/N	% (95% CI)
Cereal/w root/tub	5463/5467	99.9 (99.8, 100.0)	4879/5467	89.2 (88.4, 90.1)	5428/5467	99.3 (99.0, 99.5)
Vit A Veg	5126/5465	93.8 (93.1, 94.4)	3261/5467	59.6 (58.3, 61.0)	1357/5460	24.9 (23.7, 26.0)
D Green leafy Veg	5163/5464	94.5 (93.9, 95.1)	3333/5467	61.0 (59.7, 62.3)	1261/5460	23.1 (22.0, 24.2)
Other vegetables	5430/5465	99.4 (99.1, 99.6)	4704/5467	86.0 (85.1, 87.0)	5329/5460	97.5 (97.1, 97.9)
Vit A Fruit	4964/5449	91.1 (90.3, 91.8)	2094/5467	38.3 (37.0, 39.6)	547/5464	10.0 (9.2, 10.8)
Other fruits	5162/5466	94.4 (93.8, 95.0)	2549/5467	46.6 (45.3, 48.0)	1816/5461	33.2 (32.0, 34.5)
Meat	5130/5463	93.9 (93.2, 94.5)	1773/5467	32.4 (31.2, 33.7)	1722/5465	31.5 (30.3, 32.8)
Eggs	5203/5465	95.2 (94.6, 95.8)	2726/5467	49.9 (48.5, 51.2)	1511/5463	27.7 (26.5, 28.9)
Fish	3776/5137	73.5 (72.3, 74.7)	884/5467	16.2 (15.2, 17.2)	125/5425	2.3 (1.9, 2.7)
Leg/nut/seed	5399/5467	98.8 (98.4, 99.0)	4619/5467	84.5 (83.5, 85.4)	4643/5467	84.9 (84.0, 85.9)
Dairy	5113/5446	93.9 (93.2, 94.5)	2732/5467	50.0 (48.6, 51.3)	2503/5458	45.9 (44.5, 47.2)

Abbreviations: Cereal/w root/tub—Cereals and white roots and tubers; Vit A Veg—Vitamin A rich vegetables; D Green leafy Veg—Dark green leafy vegetables; Vit A Fruit—Vitamin A rich fruits; Leg/nut/seed—legumes, nuts and seeds; CI—Confidence Interval.

**Table 3 nutrients-12-03168-t003:** Perception of affordability and consumption of family food by socioeconomic resources, Addis Ababa, Ethiopia.

Food Groups			Wealth						Maternal Education					
	Lowest Quintile	Second Quintile	Middle Quintile	Fourth Quintile	Highest Quintile	*p* Value	Ratio Highest/Lowest	No Education	Grade 1–4	Grade 5–8	Grade 9–12	College	*p* Value	Ratio College/No Education
	(N = 1106)	(N = 1093)	(N = 1083)	(N = 1098)	(N = 1087)			(N = 752)	(N = 498)	(N = 1638)	(N = 1482)	(N = 1097)		
	%	%	%	%	%			%	%	%	%	%		
		Perceived Affordability *												
Cereal/w root/tub	81.7	88.2	90.0	91.1	95.3	<0.01	1.2	80.6	82.7	87.7	93.3	95.1	<0.01	1.2
Vit A Veg	43.7	53.8	63.3	60.6	77.3	<0.01	1.8	39.2	45.8	55.6	66.0	77.5	<0.01	2.0
D Green leafy Veg	45.3	56.2	62.3	63.5	77.8	<0.01	1.7	40.2	46.2	57.5	67.9	77.8	<0.01	1.9
Other vegetables	78.1	84.7	88.2	87.3	92.0	<0.01	1.2	76.6	76.7	84.8	90.7	92.3	<0.01	1.2
Vit A Fruit	20.3	30.2	38.2	41.7	61.4	<0.01	3.0	18.0	22.3	31.3	45.1	60.9	<0.01	3.4
Other fruits	25.5	38.8	49.2	51.5	68.5	<0.01	2.7	24.7	28.7	41.0	53.4	69.2	<0.01	2.8
Meat	16.5	25.0	29.7	34.8	56.5	<0.01	3.4	14.2	17.7	24.7	38.0	55.6	<0.01	3.9
Eggs	30.0	41.4	52.1	52.9	73.3	<0.01	2.4	26.6	33.1	44.3	57.3	71.7	<0.01	2.7
Fish	7.9	11.8	14.2	16.4	30.7	<0.01	3.9	6.9	7.2	11.2	18.7	30.5	<0.01	4.4
Leg/nut/seed	78.2	84.5	86.6	83.1	90.2	<0.01	1.2	76.6	78.7	83	88.5	89.3	<0.01	1.2
Dairy	31.7	44.1	51.1	54.9	68.5	<0.01	2.2	28.5	35.9	43.6	57.6	70.4	<0.01	2.5
		**Consumption**												
Cereal/w root/tub	99.3	99.3	99.3	99.3	99.4	0.99	1.0	99.3	98.8	99.1	99.6	99.4	0.31	1.0
Vit A Veg	16.4	17.7	25.9	26.1	38.4	<0.01	2.3	14.4	16.9	21.4	29.3	34.8	<0.01	2.4
D Green leafy Veg	16.4	20.0	23.9	23.3	32.0	<0.01	2.0	17.0	18.1	22.1	23.6	30.3	<0.01	1.8
Other vegetables	97.3	98.4	97.0	97.4	97.5	0.26	1.0	98	95.6	97.0	97.7	98.7	<0.01	1.0
Vit A Fruit	5.2	6.6	8.3	11.3	18.8	<0.01	3.6	3.5	6.8	7.0	12.0	17.9	<0.01	5.1
Other fruits	20.2	27.5	34.2	37.4	47.1	<0.01	2.3	17.6	21.7	28.0	38.6	49.8	<0.01	2.8
Meat	16.0	25.7	32.7	34.5	49.1	<0.01	3.1	17.8	16.5	27.0	35.9	48.6	<0.01	2.7
Eggs	15.7	23.2	28.0	31.5	40.2	<0.01	2.6	13.3	18.7	24.9	32.0	39.8	<0.01	3.0
Fish	0.7	1.1	1.8	2.4	5.5	<0.01	7.9	0.8	1.2	0.9	3.1	4.8	<0.01	6.0
Leg/nut/seed	88.5	87.3	84.7	83.0	81.1	<0.01	0.9	89.0	85.7	86.1	85.1	79.9	<0.01	0.9
Dairy	32.9	41.4	46.0	49.0	60.1	<0.01	1.8	28.1	31.8	41.4	54.4	59.5	<0.01	2.1

* Perceived affordability is presented as affordable for those who responded can afford to consume the food item as often as wanted. *p* value of chi-square test presented. Abbreviation: Cereal/w root/tub—Cereals and white roots and tubers; Vit A Veg—Vitamin A rich vegetables; D Green leafy Veg-—Dark green leafy vegetables; Vit A Fruit—Vitamin A rich fruits; Leg/nut/seed—legumes, nuts, and seeds.

**Table 4 nutrients-12-03168-t004:** Family food consumption and its association with perceived affordability, Addis Ababa, Ethiopia.

Family Food Groups	Model	Affordability		
		Much Less than Wanted/Not at All	Little Less than Wanted	As Much as Wanted
			OR (95% CI)	OR (95% CI)
Vit A Fruit	Unadjusted	Ref	2.56 (0.78,8.43)	3.97 (1.24,12.64)
	Adjusted ^1^	Ref	2.61 (0.77,8.75)	2.47 (0.76,8.02)
Vit A Veg	Unadjusted	Ref	1.29 (0.92,1.80)	2.35 (1.72,3.22)
	Adjusted ^1^	Ref	1.08 (0.77,1.52)	1.49 (1.08,2.07)
D Green leafy Veg	Unadjusted	Ref	1.03 (0.79,1.36)	1.78 (1.39,2.29)
	Adjusted ^1^	Ref	0.95 (0.73,1.26)	1.34 (1.02,1.74)
Other fruits	Unadjusted	Ref	1.94 (1.63,2.31)	4.11 (3.47,4.87)
	Adjusted ^1^	Ref	1.49 (1.25,1.79)	2.42 (2.00,2.92)
Meat	Unadjusted	Ref	2.46 (1.97,3.06)	5.44 (4.42,6.70)
	Adjusted ^1^	Ref	1.80 (1.43,2.26)	2.61 (2.07,3.28)
Egg	Unadjusted	Ref	2.27 (1.93,2.67)	3.27 (2.79,3.82)
	Adjusted ^1^	Ref	1.70 (1.43,2.02)	2.00 (1.67,2.40)
Dairy	Unadjusted	Ref	1.07 (0.72,1.58)	2.46 (1.72,3.54)
	Adjusted ^1^	Ref	0.93 (0.62,1.40)	1.61 (1.10,2.36)

^1^ Model adjusted for Household wealth, maternal education and food security. Clustering effect was controlled for both unadjusted and adjusted models. Abbreviation: Vit A Fruit—Vitamin A rich fruits; Vit A Veg—Vitamin A rich vegetables; D Green leafy Veg—Dark green leafy vegetables; OR—Odds ratio; CI—confidence interval; Ref—Reference group.

**Table 5 nutrients-12-03168-t005:** Family food consumption and its association with household wealth, Addis Ababa, Ethiopia.

Family Food Groups	Model	Wealth Status				
		Lowest Quintile	Second Quintile	Middle Quintile	Forth Quintile	Highest Quintile
			OR (95% CI)	OR (95% CI)	OR (95% CI)	OR (95% CI)
Vit A Fruit	Unadjusted	Ref	1.28 (0.89, 1.84)	1.66 (1.18, 2.35)	2.30 (1.65, 3.20)	4.22 (3.09, 5.77)
	Adjusted ^1^	Ref	1.09 (0.76, 1.58)	1.27 (0.89, 1.80)	1.64 (1.17, 2.30)	2.55 (1.83, 3.54)
Vit A Veg	Unadjusted	Ref	1.07 (0.85, 1.34)	1.73 (1.40, 2.14)	1.73 (1.40, 2.15)	3.08 (2.51, 3.78
	Adjusted ^1^	Ref	0.95 (0.76, 1.19)	1.42 (1.14, 1.77)	1.39 (1.11, 1.73)	2.17 (1.74, 2.69)
D Green leafy Veg	Unadjusted	Ref	1.27 (1.01, 1.58)	1.59 (1.28, 1.97)	1.54 (1.24, 1.92)	2.41 (1.96, 2.98)
	Adjusted ^1^	Ref	1.17 (0.94, 1.46)	1.37 (1.09, 1.70)	1.31 (1.04, 1.63)	1.82 (1.45, 2.27)
Other fruits	Unadjusted	Ref	1.49 (1.22, 1.82)	2.07 (1.70, 2.52)	2.33 (1.91, 2.85)	3.52 (2.89, 4.28)
	Adjusted ^1^	Ref	1.25 (1.02, 1.54)	1.51 (1.23, 1.85)	1.62 (1.31, 1.99)	1.92 (1.56, 2.37)
Meat	Unadjusted	Ref	1.81 (1.45, 2.24)	2.57 (2.08, 3.18)	2.67 (2.15, 3.3)	4.98 (4.04, 6.15)
	Adjusted ^1^	Ref	1.51 (1.21, 1.89)	1.83 (1.46, 2.29)	1.79 (1.43, 2.24)	2.66 (2.12, 3.33)
Egg	Unadjusted	Ref	1.59 (1.28, 1.98)	2.06 (1.66, 2.55)	2.43 (1.96, 3.01)	3.62 (2.93, 4.46)
	Adjusted ^1^	Ref	1.39 (1.10, 1.73)	1.58 (1.27, 1.98)	1.79 (1.44, 2.25)	2.17 (1.74, 2.72)
Dairy	Unadjusted	Ref	1.41 (1.18, 1.69)	1.76 (1.47, 2.10)	1.94 (1.62, 2.32)	3.04 (2.54, 3.65)
	Adjusted ^1^	Ref	1.24 (1.03, 1.49)	1.37 (1.14, 1.65)	1.47 (1.22, 1.77)	1.95 (1.61, 2.36)

^1^ Model adjusted for affordability, maternal education and food security. Clustering effect was controlled for both unadjusted and adjusted models. Abbreviation: Vit A Fruit—Vitamin A rich fruits; Vit A Veg—Vitamin A rich vegetables; D Green leafy Veg—Dark green leafy vegetables; OR—Odds ratio; CI—confidence interval; Ref—Reference group.

**Table 6 nutrients-12-03168-t006:** Family food consumption and its association with Maternal Education, Addis Ababa, Ethiopia.

Family Food Groups	Model	Maternal Education				
		Never Attended	Grade 1–4	Grade 5–8	Grade 9–12	College
			OR (95% CI)	OR (95% CI)	OR (95% CI)	OR (95% CI)
Vit A Fruit	Unadjusted	Ref	2.04 (1.20, 3.45)	2.10 (1.36, 3.26)	3.85 (2.52, 5.90)	6.15 (4.02, 9.41)
	Adjusted ^1^	Ref	1.97 (1.16, 3.35)	1.77 (1.13, 2.75)	2.75 (1.78, 4.25)	3.77 (2.43, 5.85)
Vit A Veg	Unadjusted	Ref	1.17 (0.86, 1.61)	1.61 (1.27, 2.04)	2.44 (1.93, 3.09)	3.06 (2.40, 3.91)
	Adjusted ^1^	Ref	1.18 (0.86, 1.62)	1.43 (1.12, 1.82)	1.88 (1.48, 2.40)	2.09 (1.62, 2.69)
D Green leafy Veg	Unadjusted	Ref	1.06 (0.78, 1.43)	1.36 (1.08, 1.70)	1.48 (1.17, 1.85)	2.09 (1.66, 2.65)
	Adjusted ^1^	Ref	1.02 (0.75, 1.38)	1.19 (0.94, 1.50)	1.15 (0.90, 1.46)	1.46 (1.14, 1.87)
Other fruits	Unadjusted	Ref	1.27 (0.95, 1.69)	1.80 (1.44, 2.24)	2.88 (2.31, 3.59)	4.51 (3.59, 5.66)
	Adjusted ^1^	Ref	1.22 (0.91, 1.64)	1.48 (1.18, 1.85)	1.97 (1.57, 2.48)	2.58 (2.03, 3.29)
Meat	Unadjusted	Ref	0.91 (0.67, 1.24)	1.70 (1.36, 2.12)	2.51 (2.01, 3.14)	4.23 (3.35, 5.33)
	Adjusted ^1^	Ref	0.85 (0.62, 1.17)	1.27 (1.01, 1.61)	1.52 (1.19, 1.92)	2.10 (1.64, 2.69)
Egg	Unadjusted	Ref	1.49 (1.09, 2.03)	2.16 (1.69, 2.75)	3.04 (2.39, 3.88)	4.25 (3.31, 5.46)
	Adjusted ^1^	Ref	1.38 (1.08, 2.04)	1.81 (1.41, 2.32)	2.11(1.64, 2.72)	2.51 (1.93, 3.26)
Dairy	Unadjusted	Ref	1.16 (0.9, 1.49)	1.77 (1.46, 2.14)	2.96 (2.44, 3.6)	3.60 (2.93, 4.43)
	Adjusted ^1^	Ref	1.13 (0.88, 1.46)	1.52 (1.25, 1.85)	2.24 (1.83, 2.73)	2.43 (1.96, 3.01)

^1^ Model adjusted for affordability, Household wealth, and food security. Clustering effect was controlled for both unadjusted and adjusted models. Abbreviation: Vit A Fruit—Vitamin A rich fruits; Vit A Veg—Vitamin A rich vegetables; D Green leafy Veg—Dark green leafy vegetables; OR—Odds ratio; CI-confidence interval; Ref—Reference group.

**Table 7 nutrients-12-03168-t007:** Mixed effect logistic regression results of family food consumption, Addis Ababa, Ethiopia.

Measure of Variation	Vit A Fruit	Vit A Veg	D Green Leafy Veg	Other Fruits	Meat	Egg	Dairy
Variance of random effect	0.10 (0.04–0.25)	0.06 (0.02–0.13)	0.08 (0.04–0.16)	0.10 (0.06–0.17)	0.24 (0.16–0.37)	0.13 (0.08–0.22)	0.07 (0.04–0.13)
SE	0.05	0.02	0.03	0.03	0.05	0.03	0.02
ICC (%)	3 (1–7)	2 (1–4)	2 (1–5)	3 (2–5)	7 (5–10)	4 (2–6)	2 (1–4)

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
