# Peer review of "The Social Stratification of Availability, Affordability, and Consumption of Food in Families with Preschoolers in Addis Ababa; The EAT Addis Study in Ethiopia"

_nutrients, 2020, doi:10.3390/nu12103168_

Round 1

Reviewer 1 Report

Thank you for the opportunity to review this study. The manuscript is generally well written. The authors sought to understand the quality of diet being consumed among families in Addis Ababa, and to what extent social stratification and perceptions of availability and affordability affect healthy food consumption. A strength of this study that contributes to its value was the large sample size and sampling design. I do have some questions that center around the sampling design used that I believe would be easily addressable by the authors. Additional details in some areas may be needed within the manuscript to help further readers and researchers. 

Line 96 - Since the survey was carried out during two different parts of the year July/August and January/February that seasonality may have impacted results?

Line 96 - Can the authors provide context as to why the survey was administered at two different time points?

Please add in details to describe how researchers ensured households who completed the July/August survey did not complete the January/February survey.

Line 106 – I’m having trouble arriving at the household sample size of 14018. If there were 116 districts * 60 households within a randomly selected cluster = 6960. If this was at each of the two-time points then that’s 13,920. Is this not correct?

Line 106 - Was a different cluster randomly selected from the remaining 4 clusters for the second round of surveying? If so, please mention this.

Line 106 – How was five chosen to break the districts into clusters?

Table 2 – Rather than showing n/N , I think it would be more clear to put the percentage however, I see that sample sizes ranged for food groups. Why is this?

Table 3- In the table legend, please report what statistical test the P-value represents

Table 3 - There's a lot of data and numbers within this table. Consider only putting the % rather than the n and % 

Line 380 – Please describe the limitations of the study.

Reviewer 2 Report

You deserve praise. However, the discussion on this topic is already sparing in expression and means, areas of extensive literature contained in the work. Final progress in the end-stage development of the plan based on the ends of the criteria for individual families according to their economic status. Such a proposal for the residents of Addis Ababa should be included in the summary.
